# Investigating the Prediction Accuracy of Recently Updated Intraocular Lens Power Formulas with Artificial Intelligence for High Myopia

**DOI:** 10.3390/jcm11164848

**Published:** 2022-08-18

**Authors:** Miki Omoto, Kaoruko Sugawara, Hidemasa Torii, Erisa Yotsukura, Sachiko Masui, Yuta Shigeno, Yasuyo Nishi, Kazuno Negishi

**Affiliations:** Department of Ophthalmology, Keio University School of Medicine, Tokyo 160-8582, Japan

**Keywords:** cataract, intraocular lens, artificial intelligence

## Abstract

The aim of this study was to investigate the prediction accuracy of intraocular lens (IOL) power formulas with artificial intelligence (AI) for high myopia. Cases of highly myopic patients (axial length [AL], >26.0 mm) undergoing uncomplicated cataract surgery with at least 1-month follow-up were included. Prediction errors, absolute errors, and percentages of eyes with prediction errors within ±0.25, ±0.50, and ±1.00 diopters (D) were compared using five formulas: Hill-RBF3.0, Kane, Barrett Universal II (BUII), Haigis, and SRK/T. Seventy eyes (mean patient age at surgery, 64.0 ± 9.0 years; mean AL, 27.8 ± 1.3 mm) were included. The prediction errors with the Hill-RBF3.0 and Kane formulas were statistically different from the BUII, Haigis, and SRK/T formulas, whereas there was not a statistically significant difference between those with the Hill-RBF3.0 and Kane. The absolute errors with the Hill-RBF3.0 and Kane formulas were smaller than that with the BUII formula, whereas there was not a statistically significant difference between the other formulas. The percentage within ±0.25 D with the Hill-RBF3.0 formula was larger than that with the BUII formula. The prediction accuracy using AI (Hill-RBF3.0 and Kane) showed excellent prediction accuracy. No significant difference was observed in the prediction accuracy between the Hill-RBF3.0 and Kane formulas.

## 1. Introduction

The primary purpose of cataract surgery is visual rehabilitation; however, refractive correction is also an important aspect of the surgery to achieve better vision and quality of life postoperatively. With modern surgical techniques, patients’ expectations for better vision postoperatively are increasing day by day. Accurately predicting the postoperative refraction in myopic eyes is challenging [1,2]. The prevalence of myopia is growing, especially in Asia [3,4,5]. While many methods are being used to overcome this issue [6,7,8,9,10,11], the challenge still remains.

The recent advances in artificial intelligence (AI) are outstanding and the application of AI in clinical medicine is a current hot topic. AI is used not only for classification or anomaly detection, but also for regression. The Hill-RBF refractive formula uses pattern recognition and data interpolation to predict postoperative refraction [12,13], whereas the Kane formula is based on theoretical optics and also incorporates regression and AI components to further refine its predictions [14]. Several studies have reported good refractive outcomes obtained with these formulas [15,16,17].

The Hill-RBF formula has been updated recently to version 3.0, which was reported to show better prediction accuracy than the previous version in a recent study; however, that study did not focus on myopic eyes (axial length [AL], 24.10 ± 1.47 mm) [13] and, to our knowledge, no other study has investigated the accuracy of the new version of the formula in myopic eyes. Therefore, we investigated the prediction accuracy of the recently updated intraocular lens (IOL) power formulas with AI for high myopia.

## 2. Materials and Methods

### 2.1. Study Institutions and Institutional Review Board Approval

The Research Ethics Committee of the Graduate School of Medicine and the Faculty of Medicine at Keio University approved this retrospective, observational study. All patients provided written consent for the surgeries. Patient consent to participate in this study was waived, and an opt-out approach was used according to the Ethical Guidelines for Medical and Health Research Involving Human Subjects presented by the Ministry of Education, Culture, Sports, Science, and Technology in Japan. The patients and public were not involved in the design, conduct, reporting, or dissemination plans of our research. This study was performed according to the tenets of the Declaration of Helsinki.

### 2.2. Participants

The study participants were retrospectively recruited at the institution. Cases of uncomplicated cataract surgery for highly myopic eyes (AL > 26.0 mm) with at least 1-month postoperative follow-up were included. Eyes with a vision-affecting ocular disease other than cataract or that had undergone past refractive surgeries were excluded. When both eyes of a patient met the criteria, we randomly selected one eye for inclusion. Seventy eyes of seventy patients were included in the final analysis. One surgeon (NK) performed all surgeries. Phacoemulsification and intraocular lens (IOL) implantation were performed through a 2.4 mm sutureless corneal incision. The implanted IOL was the TECNIS^®^ Monofocal (clear ZCB00) in 28 eyes or the yellow-tinted lens (ZCB00V) in 42 eyes (both from Johnson & Johnson, Santa Ana, CA, USA).

### 2.3. IOL Power Calculation

All patients underwent biometric measurements using the IOLMaster^®^ 700 (Carl Zeiss Meditec AG, Jena, Germany) preoperatively. Using the parameters, the postoperative refraction was predicted using five formulas: Hill-RBF3.0, Kane, Barrett Universal II (BUII), Haigis, and SRK/T. The Hill-RBF Calculator is an advanced, self-validating method for IOL power selection employing pattern recognition and a sophisticated form of data interpolation that can be used with other biconvex IOL models in the power range of +6.00 to +30.00 D and other meniscus IOL designs from −5.00 to +5.00 D [12,13]. The Kane formula is based on theoretical optics and incorporates both regression and artificial intelligence components to further refine its predictions [14]. BUII is a Gaussian-based formula for thick lenses using paraxial ray tracing, which takes into account changes in the principal plane that occur with various IOL powers, but the details are not disclosed. Hiagis and SRK/T are theoretical formulas for predicting effective lens position based on the multiple regression of the A constant, anterior chamber depth, and axial length in Haigis and the corneal curvature radius and axial length in SRK/T, respectively. The Wang Koch (WK) axial length adjustment was applied for these two formulas. [2,11,18,19]. The lens constants were set as 119.30 for the Hill-RBF3, BUII, and SRK/T formulas; 119.36 for the Kane formula; and −1.302 (A0), 0.210 (A1), and 0.251 (A2) for the Haigis formula.

### 2.4. Statistical Analysis

The best-corrected visual acuity (BCVA) measurements were performed 1 month postoperatively. First, the prediction error was calculated by subtracting the predicted value with the implanted IOL from the postoperative subjective spherical equivalent. The absolute error was then calculated as the absolute value of the prediction error. The percentages of eyes with a prediction error of ±0.25, ±0.50, and ±1.00 D were also calculated for each formula. The prediction and absolute errors were compared among the formulas using the Friedman test, followed by post hoc analysis using the pairwise Wilcoxon signed-rank test with Bonferroni correction. The percentages were compared using Cochran’s Q test, followed by post hoc analysis using the McNemar test with Bonferroni correction. Subanalyses were also performed in each of the following subgroups: eyes with an AL > 28.0 mm and eyes with an AL < 28.0 mm. Statistical significance was *p* < 0.05. All analyses were performed using R software v.4.0.4 (The R Foundation for Statistical Computing, Vienna, Austria).

## 3. Results

The demographic data of the 70 study eyes are summarized in Table 1. The mean ± standard deviation (SD) age at the time of surgery was 64.0 ± 9.0 years. The mean ± SD AL preoperatively was 27.8 ± 1.3 mm. Forty-five eyes had an AL < 28.0 mm and twenty-five eyes had an AL > 28.0 mm.

Figure 1A shows the prediction error with each formula. The values with the Hill-RBF3.0 and Kane formulas were statistically different from the BUII, Haigis, and SRK/T formulas, whereas there was not a statistically significant difference between those with the Hill-RBF3.0 and Kane formula (Table 2 and Appendix A).

Figure 1B shows the absolute error with each formula. The values with the Hill-RBF3.0 and Kane formula were smaller than that with the BUII formula, whereas there was not a statistically significant difference between the other formulas (Table 2 and Appendix A).

Figure 1C shows the stacked bar chart of the prediction accuracy with each formula. The percentage within ±0.25 D of the Hill-RBF3.0 formula was larger than that of the BUII (Table 2 and Appendix A).

Figure 2 shows the results of eyes with an AL > 28.0 mm. The prediction errors were significantly different from each other (Figure 2A, Table 3 and Appendix A). The absolute errors with the Hill-RBF3.0 and SRK/T were smaller than that with the Haigis formula (Figure 2B, Table 3 and Appendix A). The percentage within ±0.25 D with the Hill-RBF3.0 formula was larger than those with the Haigis formula (Figure 2C, Table 3 and Appendix A).

Figure 3 shows a similar analysis in the subgroup of eyes with an AL < 28.0 mm. The median prediction errors with the Hill-RBF3.0 and Kane formulas were significantly different from those with other formulas, whereas there was not a statistically significant difference between those with the Hill-RBF3.0 and Kane formulas (Figure 3A, Table 4 and Appendix A). The absolute errors with the Hill-RBF3.0 and Kane formulas were smaller than that with the BUII formula (Figure 3B, Table 4 and Appendix A). The differences in the percentages among the formulas were not significantly different from each other (Figure 3C, Table 4 and Appendix A).

## 4. Discussion

In the current study, the accuracy of the new formulas for IOL power calculations in highly myopic eyes (AL > 26.0 mm) was investigated in 70 uncomplicated cataract surgery cases. Overall, the prediction accuracy using AI (Hill-RBF3.0 and Kane) showed excellent prediction accuracy. Furthermore, this tendency was more obvious in eyes with an AL > 28.0 mm. No significant difference was observed in the prediction accuracy between the Hill-RBF3.0 and Kane formulas.

Theoretical formulas have been used for many years with continuous improvements. However, theoretical formulas are limited due to the measurement error of the AL or predicted postoperative anterior chamber depth (ACD), especially for eyes with long and short ALs outside the normal range. Recently, WK adjustment has been applied to myopic eyes [20]. The adjustment has been reported to reduce the amount of unexpected hyperopic surprise [2]. In the current study, SRK/T with WK adjustment showed excellent prediction accuracy, comparable to the new generation formulas.

Another possible approach to overcome this issue is by using AI, and the Hill-RBF and Kane formulas are representative of this approach. In the current study, these two formulas showed excellent prediction accuracy. Furthermore, this tendency was more obvious in long eyes with an AL > 28.0 mm. The Hill-RBF Calculator is a method for IOL power selection using pattern recognition and data interpolation. As more data are accumulated in the dataset, it is expected that the accuracy of this method will be further improved, and it will be able to calculate the powers for irregular cases. In fact, several negative studies have reported on the previous version of the Hill-RBF formula compared to the BUII formula [21,22]. However, in the current study, excellent prediction accuracy was observed with the updated formula. Even though there was no significant difference, only the Hill-RBF3.0 formula achieved prediction accuracy within ±0.25 D in more than half the cases in elongated eyes with an AL > 28.0 mm (Table 3). The biometric data used with the former version was AL, keratometry, and ACD, to which the central corneal thickness, lens thickness, white-to-white measurement and the sex of the patients could be added, although these were optional. These additional data were not used in the current study; however, the improved dataset with more parameters should improve the prediction accuracy of the model.

The current study has several limitations. We included a relatively small number of eyes. Although applying lens constant optimization was recommended [23], we did not do so in this study, since it was feasible only for four open-source formulas, i.e., the Haigis, Holladay 1, Hoffer Q, and SRK/T.

In conclusion, the prediction accuracy using AI (Hill-RBF3.0 and Kane) showed excellent prediction accuracy. No obvious difference was observed in the prediction accuracy between the Hill-RBF3.0 and Kane formulas.

## 5. Conclusions

This section is not mandatory but can be added to the manuscript if the discussion is unusually long or complex.

## Figures and Tables

**Figure 1 jcm-11-04848-f001:**
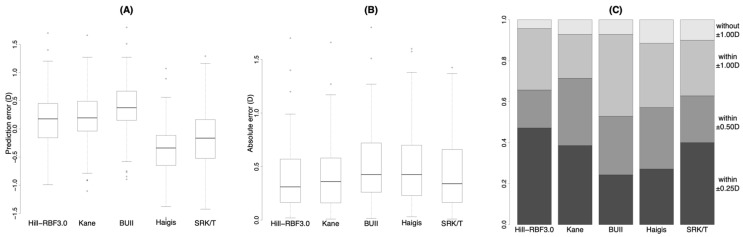
Prediction error (**A**), absolute error (**B**) and stacked bar chart (**C**) of prediction accuracy with each formula. BUII = Barrett Universal II; D = diopters. Wang Koch adjustment was applied for the Haigis and SRK/T formulas.

**Figure 2 jcm-11-04848-f002:**
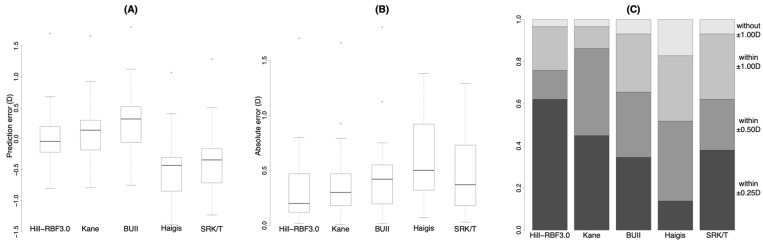
Prediction error (**A**), absolute error (**B**) and stacked bar chart (**C**) of prediction accuracy with each formula in eyes with axial length > 28.0 mm. BUII = Barrett Universal II; D = diopters. Wang Koch adjustment was applied for the Haigis and SRK/T formulas.

**Figure 3 jcm-11-04848-f003:**
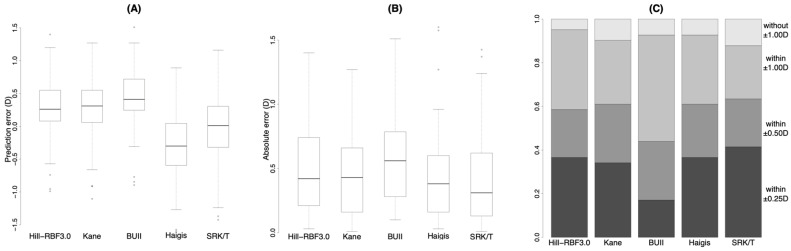
Prediction error (**A**), absolute error (**B**) and stacked bar chart (**C**) of prediction accuracy with each formula in eyes with axial length < 28.0 mm. BUII = Barrett Universal II; D = diopters. Wang Koch adjustment was applied for the Haigis and SRK/T formulas.

**Table 1 jcm-11-04848-t001:** Demographics of the study subjects.

Variables	Values
Number of eyes	70 eyes of 70 patients
Right/left	33/37
Male/female	38/32
Age at the surgery (years)	64.0 ± 9.0
Best corrected visual acuity (logMAR)	0.14 ± 0.26
Spherical equivalent (D)	−9.73 ± 4.40
Target refraction (D)	−1.79 ± 1.15
Axial length (mm)	27.84 ± 1.34
Keratometry (D)	54.3 ± 24.9
Anterior chamber depth (mm)	3.45 ± 0.35
Lens thickness (mm)	4.45 ± 0.37
Central corneal thickness (μm)	556 ± 38

logMAR = logarithm of the minimum angle of resolution; D = diopters.

**Table 2 jcm-11-04848-t002:** Prediction accuracy with each formula.

		Hill-RBF3.0	Kane	BUII	Haigis	SRK/T	*p*-Value
Prediction error (D)	Mean ± SD	0.17 ± 0.52	0.19 ± 0.51	0.36 ± 0.51	−0.38 ± 0.52	−0.18 ± 0.58	<0.001 *
	Median	0.18	0.2	0.38 ^†,‡^	−0.34 ^†,‡,§^	−0.16 ^†,‡,§,‖^	
Absolute error (D)	Mean ± SD	0.42 ± 0.34	0.42 ± 0.34	0.51 ± 0.35	0.52 ± 0.38	0.46 ± 0.38	<0.001 *
	Median	0.31	0.36	0.42 ^†‡^	0.42	0.34	
Percentage (%)	Within ±0.25 D	47.1	38.6	24.3 ^†^	27.1	40.0	0.015 *
	Within ±0.50 D	65.7	71.4	52.9	57.1	62.9	0.068
	Within ±1.00 D	95.7	92.9	92.9	88.6	90.0	0.28

* Significant difference among the formulas, calculated using Friedman test for the values and Cochran’s Q test for the percentages. ^†,‡,§,^^‖^ Significant difference from Hill-RBF 3.0, Kane, BUII, and Haigis in post hoc analysis, respectively, calculated using pairwise Wilcoxon signed-rank test for the values and the McNemar test for the percentages with Bonferroni correction. BUII = Barrett Universal II; SD = standard deviation; D = diopters. Wang Koch adjustment was applied for the Haigis and SRK/T formulas.

**Table 3 jcm-11-04848-t003:** Prediction accuracy with each formula in eyes with axial length > 28.0 mm.

		Hill-RBF3.0	Kane	BUII	Haigis	SRK/T	*p*-Value
Prediction error (D)	Mean ± SD	0.02 ± 0.48	0.10 ± 0.49	0.26 ± 0.50	−0.50 ± 0.50	−0.34 ± 0.49	<0.001 *
	Median	−0.04	0.14 ^†^	0.32 ^†,‡^	−0.43 ^†,‡,§^	−0.34 ^†,‡,§,‖^	
Absolute error (D)	Mean ± SD	0.33 ± 0.34	0.36 ± 0.34	0.43 ± 0.36	0.6 ± 0.36	0.48 ± 0.35	0.0015 *
	Median	0.20	0.30	0.42	0.50 ^†^	0.37 ^‖^	
Percentage (%)	Within ± 0.25 D	62.1	44.8	34.5	13.8 ^†^	37.9	0.0029 *
	Within ± 0.50 D	75.9	86.2	65.5	51.7	62.1	0.012 *
	Within ± 1.00 D	96.6	96.6	93.1	82.8	93.1	0.044 *

* Significant difference between the formulas, calculated using Friedman test for the values and Cochran’s Q test for the percentages. ^†,‡,§,^^‖^ Significant difference from Hill-RBF 3.0, Kane, BUII, and Haigis in post hoc analysis, respectively, calculated using pairwise Wilcoxon signed-rank test for the values and the McNemar test for the percentages with Bonferroni correction. BUII = Barrett Universal II; SD = standard deviation; D = diopters. Wang Koch adjustment was applied for the Haigis and SRK/T formulas.

**Table 4 jcm-11-04848-t004:** Prediction accuracy with each formula in eyes with axial length < 28.0 mm.

		Hill-RBF3.0	Kane	BUII	Haigis	SRK/T	*p*-Value
Prediction error (D)	Mean ± SD	0.27 ± 0.53	0.25 ± 0.52	0.42 ± 0.50	−0.3 ± 0.52	−0.06 ± 0.61	<0.001 *
	Median	0.26	0.31	0.41 ^†,‡^	−0.3 ^†,‡,§^	0.01 ^†,‡,§,‖^	
Absolute error (D)	Mean ± SD	0.49 ± 0.33	0.46 ± 0.34	0.57 ± 0.33	0.46 ± 0.38	0.45 ± 0.40	0.0017 *
	Median	0.42	0.43	0.56 ^†,‡^	0.38	0.31	
Percentage (%)	Within ±0.25 D	36.6	34.1	17.1	36.6	41.5	0.12
	Within ±0.50 D	58.5	61.0	43.9	61.0	63.4	0.22
	Within ±1.00 D	95.1	90.2	92.7	92.7	87.8	0.60

* Significant difference between the formulas, calculated using Friedman test for the values and Cochran’s Q test for the percentages. ^†,‡,§,^^‖^ Significant difference from Hill-RBF 3.0, Kane, BUII, and Haigis in post hoc analysis, respectively, calculated using pairwise Wilcoxon signed-rank test for the values and the McNemar test for the percentages with Bonferroni correction. BUII = Barrett Universal II; SD = standard deviation; D = diopters. Wang Koch adjustment was applied for the Haigis and SRK/T formulas.

## Data Availability

The data presented in this study are available on request from the corresponding author with the permission of the Keio University Ethics Committee. The data are stored, and will be discarded after the approved period by the Ethics Committee.

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
