# Peer review of "Investigating the Prediction Accuracy of Recently Updated Intraocular Lens Power Formulas with Artificial Intelligence for High Myopia"

_jcm, 2022, doi:10.3390/jcm11164848_

Round 1
Reviewer 1 Report
Nowadays, artificial-intelligence-assisted diagnosis and prediction play prominent roles in medicine. Since cataract surgery is one of the most frequently performed surgeries worldwide, the reliability of different prediction methods is of great concern. Accordingly, evaluating the prediction accuracy of intraocular lens (IOL) power is of interest to the surgical community. This paper evaluates and compares the IOL power prediction accuracy of five different formulas. The paper is well written and organized. However, I have some concerns regarding the dataset and evaluations that I list below:
1. Section 2, lines 69-70
It is mentioned that two different types of IOLs have been used in this study: Tecnis in 28 eyes and Yellow-Tinted lens in the remaining 42 lenses. However, it is reported in many studies that different types of IOLs present different postoperative behaviors that have statistically significant differences in some cases. Accordingly, I argue that judging the statistically significant difference using a dataset consisting of different IOL brands (also with different quantities) is wrong. Indeed, I believe that statistics with each formula can be biased to a particular brand of lenses.
I would suggest that the authors repeat the evaluations using sub-sets, each of which containing one particular IOL brand, and compare the statistics of each pair of formulas for different IOL brands.
Besides, I believe the current number of samples is not enough for a reliable statistical test, since these results are highly sensitive to the number of samples.
2. Tables 2-4
In these tables, the p-values for statistical differences between the formulas are reported. The p-values should be reported as the exact numbers rather than using a threshold. Besides, the authors should provide a p-value for each pair of formulas inside the paper rather than in the supplementary tables.
Author Response
Reviewer1
Nowadays, artificial-intelligence-assisted diagnosis and prediction play prominent roles in medicine. Since cataract surgery is one of the most frequently performed surgeries worldwide, the reliability of different prediction methods is of great concern. Accordingly, evaluating the prediction accuracy of intraocular lens (IOL) power is of interest to the surgical community. This paper evaluates and compares the IOL power prediction accuracy of five different formulas. The paper is well written and organized. However, I have some concerns regarding the dataset and evaluations that I list below:
Thank you for your comments. We hope this revision will meet your expectations.
- Section 2, lines 69-70
It is mentioned that two different types of IOLs have been used in this study: Tecnis in 28 eyes and Yellow-Tinted lens in the remaining 42 lenses. However, it is reported in many studies that different types of IOLs present different postoperative behaviors that have statistically significant differences in some cases. Accordingly, I argue that judging the statistically significant difference using a dataset consisting of different IOL brands (also with different quantities) is wrong. Indeed, I believe that statistics with each formula can be biased to a particular brand of lenses. I would suggest that the authors repeat the evaluations using sub-sets, each of which containing one particular IOL brand, and compare the statistics of each pair of formulas for different IOL brands. Besides, I believe the current number of samples is not enough for a reliable statistical test, since these results are highly sensitive to the number of samples.
Thank you for your comment.
According to your advice, we did the additional sub-group analysis based on the IOLs.
With ZCB00V
With ZCB00
As you can see above, almost same tendency was observed. Two types of IOLs used in this study; ZCB00 and ZCB00V, were the same brand and the same platform except for the spectral transmittance rate (colored or not), and therefore it is expected that these two IOLs would physically move and position in the same way in the eyes. Therefore, we did not include these data in the manuscript to avoid redundancy, however we will be happy to do so, if further suggested by the reviewer.
- Tables 2-4
In these tables, the p-values for statistical differences between the formulas are reported. The p-values should be reported as the exact numbers rather than using a threshold. Besides, the authors should provide a p-value for each pair of formulas inside the paper rather than in the supplementary tables.
Thank you for your comment. Some p-values were too small to be shown as the exact number, hence the current format was applied (numbers smaller than 0.001 were shown as <0.001, rather than the exact number). We believe that this format is quite common. We could not find specific instructions in this journal, however for example, NEJM and APA instruct that values smaller than 0.001 should be shown as <0.001, not the value itself, as in this manuscript (https://www.nejm.org/author-center/new-manuscripts, https://apastyle.apa.org/6th-edition-resources).
We assume that the p values in the supplemental tables were too much for readers to follow at the same time, so we put them in the supplemental materials. Instead, the significant differences were shown in the tables using some symbols as explained in the legend, hoping that readers can follow the manuscript and data more easily than otherwise.

Reviewer 2 Report
The authors investigate the prediction accuracy of intraocular lens (IOL) power formulas for high myopia. AI-based formula and traditional formulas are compared in terms of prediction accuracy. The title reflects the main subject of the manuscript and the abstract summarizes the work described in this paper. My major comments are listed below:
1) Section 2.3: IOL power calculation. Additional details are needed. Please include additional background knowledge, along with a brief description, of these methods. In addition, possible limitations and assumptions behind each formula should be specified.
2) Some of these formulas are AI-based (Hill-RBF3.0, etc.) but are not described or explained. Have they been trained on the dataset in this work? If so, how? Is the numerosity of the dataset a limitation for these methods? (overfitting, etc.).
3) Novelty: not clear the novelty of this work (if any). Also, at the end of the introduction I would suggest including a bulleted list with the highlights of the paper.
4) Tables: Some values reported in Tables 2 and 3 seem inconsistent. For example, Hill-RBF3.0 and Kane have the same mean and standard deviation for absolute error. I advise the authors to carefully check the data in Tables 2 and 3.
Author Response
Reviewer2
The authors investigate the prediction accuracy of intraocular lens (IOL) power formulas for high myopia. AI-based formula and traditional formulas are compared in terms of prediction accuracy. The title reflects the main subject of the manuscript and the abstract summarizes the work described in this paper. My major comments are listed below:
Thank you for your insightful comments. We hope this revision will be to your satisfaction.
1) Section 2.3: IOL power calculation. Additional details are needed. Please include additional background knowledge, along with a brief description, of these methods. In addition, possible limitations and assumptions behind each formula should be specified.
Thank you for your comment. Some specific description to each formula were now added in the manuscript.
2) Some of these formulas are AI-based (Hill-RBF3.0, etc.) but are not described or explained. Have they been trained on the dataset in this work? If so, how? Is the numerosity of the dataset a limitation for these methods? (overfitting, etc.).
Thank you for your comment. Hill-RBF3.0 and Kane are open-accessed tools. “We” did not train the models in this study with our datasets and we just applied the existing models to our cases. Overfitting is a condition in machine learning in which the model is fit to the training data, but not to unknown test data (in other words, not generalized). It is generally caused by a lack of data, rather than numerosity cases. We do not know how many examples were used to train the model in Hill-RBF3.0, and we don't even know if these models are simple deep learning (probably not), but we assume that the models and data sets were developed in such a way that it is not overfitting, since it is generally not that difficult to collect this type of data (cataract cases with implant of a common IOL) and the good prediction accuracies in normal cases for IOL powers from +5.00 D to -5.00 D had been already reported from other groups than the developers.
3) Novelty: not clear the novelty of this work (if any). Also, at the end of the introduction I would suggest including a bulleted list with the highlights of the paper.
Thank you for your comment. The main purpose of the study was to investigate the prediction accuracy of the latest version intraocular lens (IOL) power formulas with AI for high myopia, and novelty is that little have been investigated in other studies, as written in the introduction.
4) Tables: Some values reported in Tables 2 and 3 seem inconsistent. For example, Hill-RBF3.0 and Kane have the same mean and standard deviation for absolute error. I advise the authors to carefully check the data in Tables 2 and 3.
Thank you for your comment. We carefully re-checked the values in Table 2 and Table 3, but there were no corrections. (The consistency you indicated in Table 2 was just a coincidence.)

Round 2
Reviewer 2 Report
The authors addressed all my previous comments. The revised version of the manuscript is more clear and focused